# DensEMANN: How to Automatically Generate an Efficient while Compact DenseNet

## Abstract

We present a new and improved version of DensEMANN, an algorithm that grows small DenseNet architectures virtually from scratch while simultaneously training them on target data. Following a finite-state machine based on the network's accuracy and the evolution of its weight values, the algorithm adds and prunes dense layers and convolution filters during training only when this leads to significant accuracy improvement. We show that our improved version of DensEMANN can quickly and efficiently search for small and competitive DenseNet architectures for well-known image classification benchmarks. In half a GPU day or less, this method generates networks with under 500k parameters and between 93% and 95% accuracy on various benchmarks (CIFAR-10, Fashion-MNIST, SVHN). For CIFAR-10, we show that it comes very close to the state-of-the-art Pareto front between accuracy and size, finding networks with 98.84% of the accuracy and 98.08% of the size of the closest Pareto-optimal competitor, in only 0.70% of the search time it took to find that competitor. We also show that DensEMANN generates its networks with optimal weight values, and identify a simple mechanism that allows it to generate such optimal weights. All in all, we show this "in-supervised" essentially incremental approach to be promising for a fast design of competitive while compact convolution networks.

## 1 Introduction

The architecture of a neural network (NN) is known to have a great impact on its performance on a target task—on par with that of the training process through which the NN learns the task [1, 2, 3, 4, 5]. The main motivation behind neural architecture search (NAS) is precisely to find the most adequate architecture for a task, in the sense of achieving the highest accuracy [6, 7, 8], but also of using resources as efficiently as possible [7, 9, 8]. To this aim, NAS algorithms must compare the performance a great number of candidate architecture designs. Since naïvely training all of them from scratch would be very inefficient, the NAS community has put much effort into developing reliable and ressource-efficient performance estimation strategies [6, 7, 2, 3, 8].

In this respect, so-called "growing", "constructive" or "incremental" algorithms provide an interesting approach to NAS and performance estimation. They simultaneously build and train candidate architectures, by adding elements such as weights, neurons, layers etc. during the training process [10, 11, 12, 13]. Since new candidate networks are evaluated on basis of the weights learned by previous candidates [14, 15], the time and computation resources consumed by the entire NAS process are equivalent to those required for training a single NN [12, 14]. Furthermore, the search space is not bounded, as new elements may be added *ad infinitum* [13, 16].

This paper presents our research on DensEMANN [1], an algorithm that simultaneously grows and trains small and efficient DenseNets [17] virtually from scratch. Encouraged by previous positive

results found by its authors [1, 18], and by the great success of other similar methods [13, 19], we created a new version of this algorithm with the aim of approaching state-of-the-art performance for well-known benchmarks, or at least the state-of-the-art Pareto front between performance and model size. As a secondary goal, we tested the authors' claim that DensEMANN-generated networks perform equally or better than similar NN even when these are trained from scratch [1].

Section 2 provides some background on growing-based NAS and related research. Section 3 contains a presentation of DensEMANN's inner workings 3.1 and lists our modifications with regards to [1] 3.2. Section 4 presents our experiments and their results, and Section 5 includes our conclusions and suggestions for future research.

## 2  Rediscovery of an incremental approach

Most likely [18, 13, 20], the first growing-based NAS algorithm was Dynamic Node Creation (DNC) [21] or the cascade-correlation algorithm (CC-Alg) [22]. During the 1990's, the research field of "constructive" algorithms (as they were called) was so active that at least two contemporary surveys exist of this field [23, 24]. To our knowledge, research on growing-based NAS for convolution neural networks (CNN) only took off after the introduction of `Net2Net` operators [25] and network morphisms [26], which can instantly make a CNN wider or deeper while not changing its behaviour.

We have observed a recurring pattern of *rediscovery*, or *convergence*, between early growing techniques and more recent ones. Parallels can be made, for instance, between some network morphisms [25, 26] and early node-splitting techniques [27], or between the "backwards steps" in some modern algorithms [19, 28] and early growing-pruning hybridations [24, 23]. We believe that this convergence is helped by a direct correspondence (described in [25, 29]) between pioneering neural architectures such as multi-layer perceptrons, and more recent ones like CNN: perceptron layers correspond to convolutional layers, and perceptron neurons correspond to 3D convolution filters.

The most serious competitor to growing-based NAS algorithms are trainless or zero-cost algorithms [30, 31, 2]. These evaluate candidate NN on basis of their performance with random weights. Such methods can explore large search spaces in a matter of minutes or even seconds [31, 2]. However, extra time is still needed for training the final candidate architecture in order to use it.

## 3  DensEMANN: building a DenseNet one filter at a time

DensEMANN [1] is a growing algorithm that simultaneously builds and trains DenseNets [17] virtually from scratch. It is based on EMANN [16], an algorithm that grows multi-layer perceptrons with an analogous connection scheme to that of DenseNet, and on previous research on DenseNet-growing techniques by the same authors [18]. Based on an introspective "self-structuring" or "in-supervised" approach, much more in line with real neurology than purely performance-based NAS, it grows and prunes the candidate NN on basis of the evolution of its internal weight values.

### 3.1  General presentation of DensEMANN

By default, DensEMANN's seed architecture is a DenseNet containing a single dense block, inside which there is a single dense layer producing $k = 12$ feature maps. "Dense layers" may either be DenseNet or DenseNet-BC composite functions, with the same characteristics as in [17]. Also like in [17], the DenseNet's inputs are pre-processed by an initial convolution layer with $2 * k = 24$ filters, and its final outputs are generated through 2D batch normalization [32] and a fully connected layer.

Paralleling EMANN's "double level adaptation" [16], DensEMANN consists of two components, the macro-algorithm and the micro-algorithm [1]. Each of them builds the seed network's dense block at a different granularity level. Afterwards, the dense block may be replicated a certain user-set number of times to produce an $N$-block DenseNet (see Section 3.1.3).

### 3.1.1  The macro-algorithm

The **macro-algorithm** works at the level of dense layers, and is reminiscent of CC-Alg [22]. It iteratively stacks up dense layers in the block until there is no significant change in the accuracy (see Algorithm 1 for its pseudocode).

---
**Algorithm 1** DensEMANN macro-algorithm
---
1: **procedure** MACROALGORITHM($model$)
2:     $accuracy_{last} \leftarrow 0$
3:     $model_{last} \leftarrow model$
4:     $model, accuracy \leftarrow$ MICROALGORITHM($model$)
5:     **while** $|accuracy - accuracy_{last}| \geq IT$ **do**
6:         $accuracy_{last} \leftarrow accuracy$
7:         $model_{last} \leftarrow model$
8:         $model \leftarrow$ ADDNEWLAYER($model$)
9:         $model, accuracy \leftarrow$ MICROALGORITHM($model$)
10:     **end while**
11:     **return** $model_{last}$
12: **end procedure**
---

Each new layer is created with the same initial number of 3D convolution filters, set by a *growth rate* parameter (by default $k = 12$). In the case of DenseNet-BC, the dense layer's first convolution is created with $4 * k$ filters, and its second convolution with $k$ filters.

Before each layer addition, the macro-algorithm saves the current NN model (architecture and weights) and its accuracy. It then adds the new layer and calls the micro-algorithm to build it. Once the micro-algorithm finishes, the macro-algorithm compares the current accuracy to the one before the new layer was added. If the absolute difference between the two accuracies surpasses an *improvement threshold* (by default $IT = 0.01$), the macro-algorithm loops and creates a new layer. Otherwise, the algorithm undoes the layer's addition by loading back the last saved model, and stops there.

### 3.1.2   The micro-algorithm

The **micro-algorithm** works at the level of convolution filters. It operates only in the dense block's last layer (for DenseNet-BC, the second convolution in the last dense layer), and follows a finite-state machine with states for growing, pruning, and performance recovery.

While the network is trained through standard backpropagation, the micro-algorithm establishes different categories of filters on basis of their *kernel connection strength* (kCS). For filter $\lambda$, its kCS is the arithmetic mean of its absolute weight values $w_1, ..., w_n$.

$$kCS_\lambda = \sum_{i=1}^{n} |w_i|/n \tag{1}$$

A filter is declared "settled" if its kCS remains near-constant for the last 40 training epochs.[1] After at least $k/2$ filters have settled, settled filters can be declared "useful" if their average kCS over the last 10 epochs falls above a *usefulness threshold* (UFT), and "useless" if that same value falls below a *uselessness threshold* (ULT). During the micro-algorithm's improvement stage (1), the UFT and ULT are recalculated after each training epoch on basis of the maximum and minimum kCS among settled filters, and of user-settable parameters $UFT_{auto}$ and $ULT_{auto}$ (by default respectively 0.8 and 0.2):

$$UFT = UFT_{auto} * \left( \max_{\lambda\, is\, settled}(kCS_\lambda) - \min_{\lambda\, is\, settled}(kCS_\lambda) \right) + \min_{\lambda\, is\, settled}(kCS_\lambda) \tag{2}$$

$$ULT = ULT_{auto} * \left( \max_{\lambda\, is\, settled}(kCS_\lambda) - \min_{\lambda\, is\, settled}(kCS_\lambda) \right) + \min_{\lambda\, is\, settled}(kCS_\lambda) \tag{3}$$

The micro-algorithm uses these three filter categories—settled, useful and useless—as references for building the network's last layer. To do this, it alternates between three stages (see Figure 1):

1. **Improvement**: the network starts training with initial learning rate $LR_0 = 0.1$, and filters are progressively declared settled, then useful or useless. Meanwhile, a countdown begins with a fixed length in training epochs: the *patience parameter* (by default $PP = 40$ epochs).

---
[1]The actual criterion is: if the first derivative of the kCS, calculated as the difference between the current kCS and the one 10 epochs ago divided by 10-1=9, remains near-zero for the last 30 epochs.

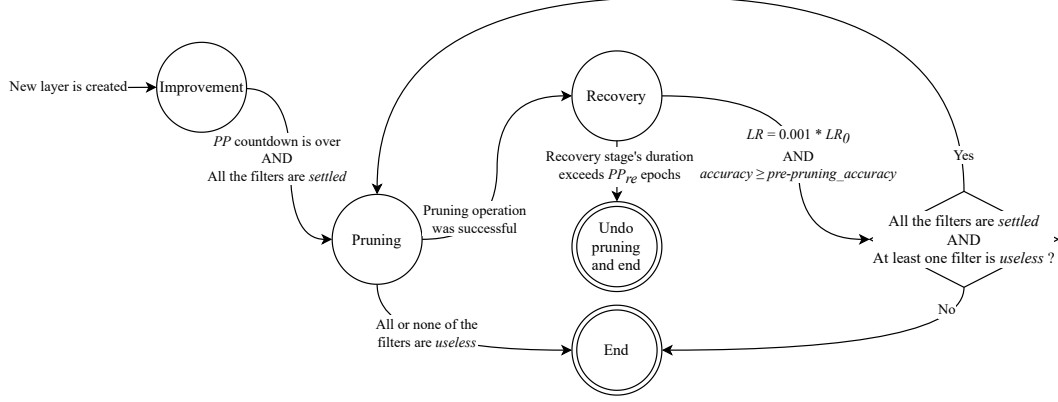

Figure 1: Flowchart of the finite-state machine for DensEMANN's micro-algorithm.

The learning rate (LR) is divided by 10 after 50% and 75% of this countdown has elapsed. If at any point during this stage the number of useful filters exceeds its last maximum value, a new filter is added and the countdown and LR are reset. The stage ends when (1) the countdown is over and (2) all filters have settled. The next stage is always 2 (pruning).

2. **Pruning**: if all or none of the layer's filters are useless, the micro-algorithm ends here. Otherwise, the micro-algorithm saves the NN model and its accuracy (like the macro-algorithm does before creating a new layer), deletes all useless filters, and moves to stage 3 (recovery). From the first pruning stage onwards, the UFT and ULT values are frozen.

3. **Recovery**: the network is trained again, with the same $PP$-epoch countdown and the same initial and scheduled LR values as in stage 1 (improvement), but without filter additions. There are two additional countdowns, one with the same length as the last improvement stage ($PP$ + the number of epochs it took for all filters to settle), and another one with a length of $PP_{re} > PP$ epochs (by default $PP_{re} = 130$). Three things may happen:

   (a) If (1) the learning rate has already reached its lowest scheduled value (i.e. in practice after $0.75 * PP$ epochs) and (2) the current accuracy has reached or surpassed its pre-pruning value, the stage ends. If at this point (3) all the filters have settled and (4) there is at least one useless filter, the next stage is 2 (pruning). Otherwise, the micro-algorithm ends.

   (b) If the stage's duration exceeds $PP_{re}$ epochs, the previous pruning operation is considered "fruitless" and undone. The pre-pruning model is loaded back, and the micro-algorithm ends.

   (c) If the stage's duration exceeds that of the previous improvement stage, the filters' kCS values are considered "frozen". In practice, this means that all filters are declared settled at most 40 epochs after this point, and that the frozen kCS values will be used as the reference for any subsequent pruning.

DensEMANN's weight initialization mechanisms are also worth commenting:

1. The weights for new layers are initialized along a truncated normal distribution, similar to that of TensorFlow v1's [33] "variance scaling initializer". For this initializer, the distribution's standard deviation (SD) is usually inversely proportional to each layer's number of input features and to its filters' dimensions. However, for layers that the micro-algorithm can act upon, the distribution's SD only depends on the filters' dimensions, resulting in bigger initial weights in these layers.

2. The weights of new filters are initialized using a special *complementarity* mechanism borrowed from EMANN [16]. The weights' absolute values are random, and follow the same truncated normal distribution as weights in new modifiable layers. Their sign configuration, however, is not random: it is the inverted sign configuration of the filter with the lowest kCS. This is done to ensure the mutual complementarity and co-adaptation of new and old filters.

### 3.1.3 Building more than one dense block

DensEMANN can also be set to build DenseNets with a user-set number of dense blocks $N$. To do this, DensEMANN first uses the macro- and micro-algorithms to build a one-block DenseNet, and then replicates the generated dense block $N-1$ times to create a $N$-block DenseNet.

Between blocks, transition layers are created with a similar architecture to that in [17], i.e. with a batch normalisation [32], a ReLU function [34], a 1x1 convolution and a 2x2 average pooling layer [35]. The number of filters in the 1x1 convolution depends on whether the network is a DenseNet or a DenseNet-BC: for DenseNet it is the same as the previous block's number of output features, while for DenseNet-BC it is multiplied by a reduction factor, by default $\theta = 0.5$.

The weights for the $N-1$ new blocks are initialized using the same method that DensEMANN uses for new layers.[2] After the new blocks are added, the NN is trained for 300 extra epochs. The LR recovers its initial value $LR_0$ at the beginning of these last 300 epochs, and is divided by 10 on epochs 150 and 255 (i.e., 50% and 75% through the extra training epochs). During these epochs, DensEMANN adopts a "best model saving" approach: the NN's weights are saved whenever its loss reaches a new minimum value, and after the 300 epochs, these "best" weights are loaded back to allow the NN to reach optimal performance.

Although it is in direct contrast with DensEMANN's incremental philosophy, this method for replicating the generated block $N-1$ times is activated by default, with $N=3$. Its development was motivated by previous experimental results with mechanisms that copy DensEMANN-generated layers a predefined number of times, and by the good performance of cell-based NAS approaches [13, 14, 36] that first search for a small neural pattern (the cell) and then replicate it $N$ times. In Appendix A, we give the results of an ablation study that compares, among others, DensEMANN's performance with and without this dense block replication mechanism.

### 3.2 Differences with the original DensEMANN

Below are the differences between our version of the algorithm and the one described in [1]:

1. Changes to the macro-algorithm:
   (a) The last layer addition is always undone, as it does not fulfil the accuracy improvement criterion. This was suggested in [1], and EMANN uses a similar mechanism [16].
   (b) The improvement threshold's default value was changed to $IT = 0.01$. Observations in [1] suggest that, with the previous default value (0.005), the last few layer additions do not have a big impact in the NN's final accuracy.
2. Changes to the micro-algorithm:
   (a) Only settled filters may be declared useful or useless. This was proposed in [1] as a means to avoid quick cascades of often superfluous filter additions.
   (b) The patience parameter's default value was changed to $PP = 40$ epochs. Observations in [1] suggest that it takes approximately 40 epochs for all the filters in a layer to settle.
   (c) If *at any point* during the improvement stage the number of useful filters exceeds its last highest value, a new filter is added and the patience countdown is reset. In v1.0, this can only happen if the countdown has not yet ended.
   (d) The pruning and recovery stages have been heavily modified to avoid long recovery stages and their effects. We indeed observed that the kCS of settled filters is not constant but actually decreases very slowly over time. If the recovery stage is too long, this causes a very harsh pruning after which the accuracy cannot be recovered.
3. We added a method that replicates the generated dense block $N$ times.

## 4 Experiments

We re-implemented DensEMANN from scratch using the PyTorch (v1.8 or higher[3]) [37] and Fastai (v2.5.3) [38] Python libraries. Our code is based on the DenseNet implementation for PyTorch by

---

[2]Except for transition layers and the first layers in each block, whose weights are initialized with zero values.

[3]Depending on the computational environment, we used PyTorch v1.8.1 or v1.10.0+cu113 for running our experiments. See further below.

Pleiss et al. [39] and on the original DensEMANN implementation by García-Díaz [40] (both under MIT license). We initially replicated the latter as faithfully as possible, including the unusual weight initialization described in Section 3.1.2. Then, we made the modifications described in this paper.

In our experiments we use three well-known image classification benchmarks: CIFAR-10 [41], Fashion-MNIST [42] and SVHN [43]. For each training process (either when running DensEMANN or training NN from scratch), we split the target data into three parts:

- Training set: a random set of examples, different for each training process. It is used for training the NN. For CIFAR-10 and Fashion-MNIST it contains 45,000 "training" images, and for SVHN it contains 6,000 images from the "train" and "extra" sets.

- Validation set: another random set of examples, different for each training process but separate from the training set (there is no overlap between the two sets). It is used for estimating the NN's accuracy and loss during training (and in the case of DensEMANN, during growing). For CIFAR-10 and Fashion-MNIST it contains 5,000 "training" images, and for SVHN it contains 6,000 images from the "train" and "extra" sets.

- Test set: a predefined set of examples that the NN never "sees" during training. It is used for evaluating the NN's final performance (accuracy and cross-entropy loss), and to compare it against the state of the art. It is the entire set of "test" images provided by each dataset's authors: 10,000 images for CIFAR-10 and Fashion-MNIST, and 26,032 images for SVHN.

A batch size of 64 images is used for all datasets, and for all three of the above splits.

Our data pre-processing workflow is as follows:

1. Random crop with 4-pixel padding + random horizontal flip (as in [17]), only for CIFAR-10's training and validation data.

2. Normalization. For CIFAR-10 we use the dataset's channel-wise mean and SD values as in [17]. For the other two datasets we assume mean and SD values of 0.5 for all channels.

3. Cutout regularization [44], only for the training and validation data.

DensEMANN's parameters were set to their default values: $IT = 0.01$, $PP = 40$, $PP_{re} = 130$, $LR_0 = 0.1$, $k = 12$ for the first layer, $N = 3$ blocks in the final network. All other default values are the same as in [1]. We opted to generate DenseNet-BC architectures, as in past research they provided better results than standard DenseNet [1, 18, 17].

For our experiments, we used the following computation environments:

- MSi GT76 Titan DT laptop: Windows 10 Pro (64-bit) OS, Intel Core i9-10900K CPU (3.70 GHz), NVIDIA GeForce RTX 2080 Super GPU, 64.0 GB RAM (63.9 GB usable). Python is v3.9.8, PyTorch is v1.10.0+cu113.

- Internal cluster: Linux Ubuntu 20.04.4 LTS (x86-64) OS, 16 AMD EPYC-Rome Processor CPUs (2.35 GHz), NVIDIA GeForce RTX 3090 GPU, 64 GB RAM. Python is v3.8.6, PyTorch is v1.8.1.

In Table 1, GPU times in black were obtained with the MSi GT76, while GPU times in *italized purple* were obtained on the internal cluster. We consider the times obtained on the MSi GT76 to be more reliable, as on the internal cluster we have let up to four tests run at a time, whereas on the MSi GT76 we have only run one test at a time.

The total computation time for all the experiments in this paper (excluding the appendices) was 6.49 GPU days (3.61 days on the MSi GT76 and 2.88 days on the internal cluster). Below are the computation times for each experiment:

- 4.1: 4.88 GPU days (2.81 on the MSi GT76, 2.07 on the cluster).

- 4.2: 1.61 GPU days (0.80 on the MSi GT76, 0.80 on the cluster).

## 4.1 DensEMANN's full potential unlocked

We began by running DensEMANN 5 times for each dataset, in order to get an idea of the algorithm's performance. The results of this experiment correspond to the two first lines for each dataset in

Table 1: Using DensEMANN for growing and training DenseNet-BC on benchmark datasets

| Dataset | Experiment | GPU execution time (hours) | GPU inference time (seconds) | Num. layers per block | Trainable parameters (k) | Validation set | | Test set | |
|---|---|---|---|---|---|---|---|---|---|
| | | | | | | Acc. (%) | Loss | Acc. (%) | Loss |
| CIFAR-10 | Average performance | 13.48 ± 2.72 | 3.21 ± 0.36 | 5.8 ± 1.6 | 186.36 ± 56.68 | 90.06 ± 1.38 | 0.30 ± 0.04 | 93.41 ± 0.90 | 0.23 ± 0.03 |
| | Best network | 16.55 (67.39) | 3.47 | 7 | 245.42 | 91.34 | 0.26 | 93.91 | 0.21 |
| | Best network retrained | 3.86 ± 0.01 | 3.46 ± 0.04 | 7 | 245.42 | 91.90 ± 0.42 | 0.25 ± 0.01 | 94.25 ± 0.16 | 0.20 ± 0.01 |
| Fashion-MNIST | Average performance | *6.55 ± 1.80* | *3.98 ± 0.35* | 2.2 ± 1.3 | 51.84 ± 25.51 | 92.63 ± 0.73 | 0.20 ± 0.02 | 93.68 ± 0.68 | 0.20 ± 0.01 |
| | Best network | *7.53 (32.75)* | *4.26* | 3 | 68.64 | 93.62 | 0.18 | 94.43 | 0.19 |
| | Best network retrained | *2.81 ± 0.02* | *3.75 ± 0.20* | 3 | 68.64 | 93.70 ± 0.53 | 0.18 ± 0.01 | 94.47 ± 0.22 | 0.19 ± 0.01 |
| SVHN | Average performance | *3.39 ± 0.26* | *13.36 ± 0.33* | 11.0 ± 1.2 | 339.81 ± 63.39 | 93.38 ± 0.47 | 0.24 ± 0.02 | 94.43 ± 0.29 | 0.27 ± 0.02 |
| | Best network | *3.23 (16.96)* | *13.28* | 11 | 336.07 | 94.10 | 0.22 | 94.70 | 0.26 |
| | Best network retrained | *1.04 ± 0.17* | *11.76 ± 2.69* | 11 | 336.07 | 93.81 ± 0.39 | 0.23 ± 0.01 | 94.50 ± 0.16 | 0.26 ± 0.01 |

Table 1: the "average performance" lines contains the mean and SD over the 5 runs for each variable, whereas the "best network" line corresponds to the DensEMANN-generated NN that obtained the lowest (cross-entropy) loss on the validation set. In the latter case, we indicate two execution times: the execution time for the run that generated this "best" NN, and the total execution time for all 5 runs of DensEMANN (i.e. the total GPU time that we consumed to search for this optimal candidate).

All in all, DensEMANN performs very well on all three benchmarks. The generated architectures are always under 0.5 million parameters (in the case of Fashion-MNIST they are even under 0.1 million parameters), yet the average test set accuracies are all between 93% and 95%. The current state of the art test set accuracies on CIFAR-10 [45] and SVHN [46] are at 99% or higher, while that on Fashion-MNIST [47] is at just under 97%. This said, the top-performing models for these benchmarks are very large, containing several millions of parameters.

Concerning DensEMANN's execution times, they range from around 3 hours (SVHN) to just over half a day (CIFAR-10). Consequently, it always took us less than 3 days to run DensEMANN 5 times, and find our best candidate network for all benchmarks.

It is noteworthy that DensEMANN *does* seem to build minimal architectures that adapt to each dataset's peculiarities. For Fashion-MNIST, a grayscale dataset with smaller images than the other two datasets, DensEMANN generated very small and shallow architectures—an order of magnitude smaller than those for the two other datasets. Meanwhile, the biggest and deepest NN were generated for SVHN, an RGB dataset whose images contain distractors around the main data to classify.

## 4.2 Retraining our best networks from scratch: DensEMANN vs. "perfect" NAS

For our second experiment, we erased the weights in DensEMANN's best network for each dataset, replaced them with randomly initialized weights (using an exact copy of TensorFlow's "variance scaling initializer") and trained the network from scratch 5 times for 300 epochs. We used the same workflow as for the block replication mechanism at the end of DensEMANN: beginning with a LR value of $LR_0 = 0.1$, we divide it by 10 on epochs 150 and 255. We also use the same "best model saving" approach, where we save the weight values that produce the lowest validation set loss, and load them back at the end of the training process. The results of this experiment correspond to the "best network retrained" lines in Table 1.

This experiment is useful for two reasons. On one hand, it allows us to test the claims in [1] that DensEMANN generates its networks with optimal weights. If that were the case, the network's performance (accuracy and loss) with the original and retrained weights would not be very different. On the other hand, it can be used for comparing DensEMANN's execution times to those of a hypothetical "perfect" NAS algorithm. As explained in Section 2, even if we imagine a "perfect" zero-cost NAS algorithm that can instantly find an optimal network in DensEMANN's search space, one still needs to train the candidate network before using it. For this reason, we consider the GPU time cost of retraining DensEMANN's final architecture to be a good proxy for the GPU time cost of using such a "perfect" NAS algorithm to explore DensEMANN's search space.

Concerning the network's performance, the difference before and after being retraining is not very big, and this is true on both the validation and test set. In the case of CIFAR-10, there is an improvement for the accuracy, but the loss does not seem significantly different. One-sample T-tests confirm this observation: the only statistically significant differences are found for CIFAR-10's validation and test set accuracies (respectively $P = 0.039$ and $P = 0.010$), and for SVHN's test set accuracy ($P = 0.044$). In the former case the accuracy improved after retraining, while in the latter case it

worsened after retraining. Nevertheless, in both cases the test loss does not change significantly, which leads us to conclude that DensEMANN *does* optimally train the architectures that it grows.

Concerning the GPU times, for all datasets there is a significant difference between DensEMANN's execution times (both the average time and that of the best run) and the average time cost for retraining the best run's final candidate NN. The mean GPU time cost of DensEMANN is 3.49 times longer than that of retraining for CIFAR-10, 2.33 times longer for Fashion-MNIST, and 3.25 times longer for SVHN. Part of this difference most certainly comes from the 300 extra training epochs required by DensEMANN's block replication mechanism—exactly the same number of epochs that we use to retrain the best generated NN from scratch. Since for two similar architectures 300 training epochs will represent a similar GPU time, it is mathematically impossible for DensEMANN to outperform "perfect" NAS' time cost when using block replication. Nevertheless, the GPU time costs of DensEMANN and SGD training remain of the same order of magnitude (hours).

### 4.3 Comparison against the state of the art

In Table 2 and Figure 2, we take advantage of the widespread use of CIFAR-10 as a benchmark task to map the state of the art for the compromise between model size and error rate on this dataset, and see where DensEMANN fits with regards to that state of the art. Concretely, we use Figure 2 to visualize the current Pareto front for the size vs. error rate compromise. We compare DensEMANN to this Pareto front by identifying the NN models and NAS algorithms that make up this front, and by focusing on the ones that are closest to DensEMANN's performance.

The closest Pareto-optimal competitor to DensEMANN is LEMONADE S-I [19], another growing-based algorithm that is designed specifically to explore the Pareto front between model acccuracy and size. After 80 GPU days LEMONADE S-I generated a final candidate architecture with 190 thousand parameters that obtained 94.5% accuracy on CIFAR-10. DensE-MANN's average performance is very close to this: with 98.08% of its size, we reach 98.84% of the LEMONADE S-I network's accuracy, in 0.70% of the GPU time that LEMONADE took to find that network.

Concerning the execution time, a closer competitor to DensE-MANN is NASH Random, by

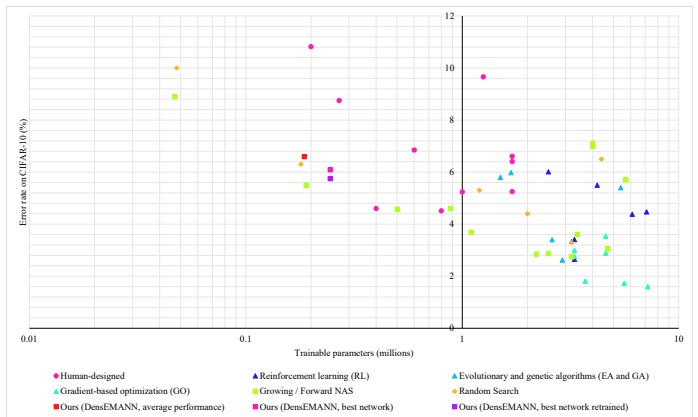

Figure 2: Scatter plot of the performance of the human-designed NN models and NAS algorithms in Table 2 (including DensE-MANN): accuracy on CIFAR-10 vs. size in trainable parameters.

the same authors as LEMONADE [14]. On average, it takes 0.19 GPU days to produce its final candidate networks, their average size is 4.4 million parameters, and their average accuracy on CIFAR-10 is 93.50%. DensEMANN reaches a similar accuracy with 4.24% of the size, but its average execution time is 2.96 times longer. It is likely that network morphisms [25, 26], which NASH uses for preserving the NN's behaviour after growing, are in part responsible for its quick execution time.

## 5 Conclusions and future work

We present DensEMANN, an "in-supervised" growing-based NAS algorithm that simultaneously builds and trains DenseNet architectures for target tasks. We show that, in half a GPU day or less, DensEMANN can generate very small networks (under 500 thousand trainable parameters) for various benchmark image classification tasks, training them with optimal weights that allow them to reach around 94% accuracy on these benchmarks. For one of them (CIFAR-10), we show that

Table 2: Performance comparison of DensEMANN against human-designed NN models and state-of-the-art NAS algorithms, for architectures with less than 10 million parameters

| Category | Name | Trainable parameters (M) | Error rate on CIFAR-10 (%) | GPU execution time (days) |
|---|---|---|---|---|
| Human-designed | ResNet 20 [48] | 0.27 | 8.75 | N/A |
| | ResNet 110 (as reported by He et al. [48]) | 1.7 | 6.61 ± 0.16 | N/A |
| | ResNet 110 (as reported by Huang et al. [49]) | 1.7 | 6.41 | N/A |
| | ResNet 110 with Stochastic Depth [49] | 1.7 | 5.25 | N/A |
| | WRN 40-1 (no data augmentation) [50] | 0.6 | 6.85 | N/A |
| | DenseNet 40 (k = 12) [17] | 1 | 5.24 | N/A |
| | DenseNet-BC 100 (k = 12) [17] | 0.8 | 4.51 | N/A |
| | Highway 1 (Fitnet 1) [51] | 0.2 | 10.82 | N/A |
| | Highway 4 [51] | 1.25 | 9.66 | N/A |
| | Petridish initial model (N=6, F=32) + cutout [13] | 0.4 | 4.6 | N/A |
| Reinforcement learning (RL) | NAS-RL / REINFORCE (v1 no stride or pooling) [52] | 4.2 | 5.5 | 22400 |
| | NAS-RL / REINFORCE (v2 predicting strides) [52] | 2.5 | 6.01 | 22400 |
| | NAS-RL / REINFORCE (v3 max pooling) [52] | 7.1 | 4.47 | 22400 |
| | NASNet-A (6 @ 768) [36] | 3.3 | 3.41 | 2000 |
| | NASNet-A (6 @ 768) + cutout [36] | 3.3 | 2.65 | 2000 |
| | Block-QNN-S, N=2 [53] | 6.1 | 4.38 | 96 |
| Evolutionary and genetic algorithms (EA and GA) | Large-Scale Evolution [54] | 5.4 | 5.4 | 2600 |
| | CGP-CNN (ConvNet) [55] | 1.5 | 5.8 | 12 |
| | CGP-CNN (ResNet) [55] | 1.68 | 5.98 | 14.9 |
| | AmoebaNet-A (N=6, F=32) [56] | 2.6 | 3.4 ± 0.08 | 3150 |
| | AmoebaNet-A (N=6, F=36) [56] | 3.2 | 3.34 ± 0.06 | 3150 |
| | EcoNAS + cutout [57] | 2.9 | 2.62 ± 0.02 | 8 |
| Gradient-based optimization (GO) | ENAS + micro search space [58] | 4.6 | 3.54 | 0.45 |
| | ENAS + micro search space + cutout [58] | 4.6 | 2.89 | 0.45 |
| | DARTS (1st order) + cutout [59] | 3.3 | 3 ± 0.14 | 1.5 |
| | DARTS (2nd order) + cutout [59] | 3.3 | 2.76 ± 0.09 | 4 |
| | XNAS-Small + cutout [60] | 3.7 | 1.81 | 0.3 |
| | XNAS-Medium + cutout [60] | 5.6 | 1.73 | 0.3 |
| | XNAS-Large + cutout [60] | 7.2 | 1.6 | 0.3 |
| Growing / Forward NAS | NASH ($n_s teps = 5$, $n_n eigh = 8$, 10 runs) [14] | 5.7 | 5.7 ± 0.35 | 0.5 |
| | LEMONADE SS-I + mixup + cutout [19] | 0.047–3.4 | 8.9–3.6 | 80 |
| | LEMONADE SS-II + mixup + cutout [19] | 0.5–13.1 | 4.57–2.58 | 80 |
| | Petridish macro (N=6, F=32) + cutout [13] | 2.2 | 2.85 ± 0.12 | 5 |
| | Petridish cell (N=6, F=32) + cutout [13] | 2.5 | 2.87 ± 0.13 | 5 |
| | Petridish cell, more filters (N=6, F=37) + cutout [13] | 3.2 | 2.75 ± 0.21 | 5 |
| | Firefly, WRN 28-1 seed + BN [10] | 4 | 7.1 ± 0.1 | N/A |
| | GradMax, WRN 28-1 seed + BN [10] | 4 | 7.0 ± 0.1 | N/A |
| Random Search | DARTS Random + cutout [59] | 3.2 | 3.29 ± 0.15 | 4 |
| | NASH Random ($n_s teps = 5$, $n_n eigh = 1$) [14] | 4.4 | 6.5 ± 0.76 | 0.19 |
| | LEMONADE SS-I Random + mixup + cutout [19] | 0.048–2 | 10–4.4 | 80 |
| Ours | DensEMANN (average performance) + cutout | 0.056 ± 0.009 | 13.91 ± 1.28 | 0.33 ± 0.05 |
| | DensEMANN (best network) + cutout | 0.245 | 6.09 | 2.81 |
| | DensEMANN (best network retrained) + cutout | 0.245 | 5.75 ± 0.16 | 2.97 ± 0.00 |

DensEMANN's performance is very close to state-of-the-art Pareto-optimality for the compromise between accuracy and neural architecture size. This said, by studying DensEMANN in detail and comparing it to other algorithms in the literature, we find methodologies—such as network morphisms and best model saving—that could make this approach even quicker and more optimal.

Future research on improving DensEMANN could follow some of the following research lines:

- Developing a zero-cost NAS algorithm that quickly explores DensEMANN's search space. This should become the baseline for evaluating future DensEMANN-inspired algorithms.

- Further incorporating best model saving into DensEMANN: during the micro-algorithm's improvement and recovery stages, save the weights—and the model—that correspond to the best validation loss since the start of the stage, then reload them at the end of the stage. This could double as a method for controlling the true usefulness of filter additions.

- Comparing different ways to initialize new filters or layers, such as network morphisms [25, 26] and GradMax [10].

- Replacing the block replication method with an improved macro-algorithm, that can decide when it is more convenient to start a new block or to just add a new layer in the current one.

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
