# OpenReview forum: "DensEMANN: How to Automatically Generate an Efficient while Compact DenseNet"
_NeurIPS.cc/2023/Conference — Submitted to NeurIPS 2023_

### Official Review · Reviewer_Jgdu · 2023-07-04

**Soundness:** 2 fair
**Presentation:** 2 fair
**Contribution:** 2 fair
**Rating:** 4
**Confidence:** 4

**Summary:**

In this paper, the authors propose an enhanced version of DensEMANN, which efficiently grows and trains small DenseNet architectures. They employ a macro-algorithm to expand new layers and utilize a micro-algorithm to construct new convolution operations. Through iterative layer growth, this method generates novel architectures within a few GPU hours.

**Strengths:**

- Detailed experimental settings are provided in this paper.

**Weaknesses:**

- The motivation behind this research requires additional clarification.
- As shown in Table 2, GO methods consumes the leaset GPU days and achieves the best performance. Therefore, it raises the question of why not directly utilize the GO methods.
- The experimental comparison with the original DensEMANN is missing.
- Experiments are only conducted on small datasets. Can DensEMANN be applied on large datasets, for example ImageNet-1k?
- Too many hyper-parameters are introduced in this method, which brings difficulties for applying this method on other tasks.

**Questions:**

- By automatically growing small DenseNet architectures, what insights can we obtain in architectural design?

**Limitations:**

Please see the Weaknesses.

---

> ### Author Rebuttal · Authors · 2023-08-10
>
> > The motivation behind this research requires additional clarification.
>
> Our main motivation was to improve upon DensEMANN in order to make it reach its "full potential". More specifically, as stated in the introduction, we designed "a new version of this algorithm with the aim of approaching [i.e. getting as close as possible to] state-of-the-art performance for well-known benchmarks, or at least the state-of-the-art Pareto front between performance and model size".
>
> We also had a secondary goal of testing the claim in (García-Díaz and Bersini 2021) that DensEMANN-generated NN reach equal or better performance to similar NN when these are trained from scratch. That paper only provided "reliable" results for DensEMANN's micro-algorithm, as the full method was still very unstable. With an improved and stabilized DensEMANN, we were able to establish more reliable comparisons with a variety of baselines, including retraining the DensEMANN-generated NN from scratch (Section 4.2 and Appendix A.1.2), and comparing DensEMANN's performance with a naive NAS algorithm that focused only on DenseNet (Appendix A.2).
>
>
> > As shown in Table 2, GO methods consumes the leaset GPU days and achieves the best performance. Therefore, it raises the question of why not directly utilize the GO methods.
>
> Various reasons:
> * The difference in GPU days between GO methods and DensEMANN (and similar growing-based algorithms such as NASH) is not that large. Both are usually at around 0.5 or 0.3 GPU days. The cause for this is likely that the two methods' basic NAS paradigms are very similar: both GO and growing-based methods look for optimal modifications to an architecture while it is being trained.
> * Table 2 indeed shows that the performance of GO methods is greater than DensEMANN's, but at the cost of bigger final models (in terms of trainable parameters). Even if they were tweaked to generate small models like DensEMANN's, GO methods are often based on extracting subnetworks out of overparametrized supernetworks, which themselves are often very large.
> * GO methods such as ENAS and DARTS have been criticised in the past for various reasons (see Fair DARTS, Chu et al. 2020). Their main drawbacks are the unfair advantages that supernetwork structures give to certain architecture elements (mainly skip connections in the case of DARTS), and their inability to properly represent a continuous encoding of discrete subnetwork choices (the architecture weights associated with a choice between different elements are often too close together to distinguish which choice is "best").
>
> This said, we _would_ like to implement GO-based functionalities in future versions of DensEMANN, mainly to broaden the search space. For instance, the dense block topology could be interpreted as a supernetwork, and architecture weights could be attributed to connections between layers in order to extract an optimal non-dense subnetwork. We are also interested in GO-based methods that allow for a choice between different kernel sizes, such as superkernels (Stamoulis et al. 2019).
>
>
> > The experimental comparison with the original DensEMANN is missing.
>
> The original DensEMANN was too unstable for an experimental comparison against the full algorithm (macro- and micro-algorithm). However, we were able to run a comparison in terms of the micro-algorithm alone. The below table compares the results from the original paper (Table I of García-Díaz and Bersini 2021) to our own for the same experiment (means and standard deviations over 5 runs).
>
> | DensEMANN version | Dataset | Architecture style | GPU execution time (hours) | Num. output filters in layer | Trainable parameters (k) | Test set acc. (%) |
> |:---:|:---:|:---:|:---:|:---:|:---:|:---:|
> | Original (García-Díaz and Bersini, 2021) | CIFAR-10 | DenseNet | 1.66 ± 1.06 | 11.40 ± 2.07 | 3.6 ± 0.5 | 62.74 ± 2.92 |
> |  |  | DenseNet-BC | 1.90 ± 0.16 | 21.60 ± 0.55 | 11.8 ± 0.2 | 71.61 ± 1.11 |
> |  | SVHN | DenseNet | 0.26 ± 0.01 | 13.40 ± 0.55 | 4.1 ± 0.1 | 47.89 ± 3.10 |
> |  |  | DenseNet-BC | 0.45 ± 0.12 | 17.00 ± 3.16 | 9.8 ± 1.4 | 57.81 ± 1.08 |
> | Ours | CIFAR-10 | DenseNet | 1.59 ± 0.51 | 13.80 ± 1.79 | 4.1 ± 0.4 | 63.17 ± 0.76 |
> |  |  | DenseNet-BC | 1.30 ± 0.32 | 13.20 ± 1.10 | 8.1 ± 0.5 | 69.48 ± 0.78 |
> |  | SVHN | DenseNet | 0.17 ± 0.04 | 10.60 ± 2.88 | 3.4 ± 0.7 | 38.59 ± 2.63 |
> |  |  | DenseNet-BC | 0.15 ± 0.00 | 10.00 ± 0.71 | 6.7 ± 0.3 | 45.40 ± 0.36 |
>
> This table shows that our version of DensEMANN produces smaller layers than the original algorithm, but does so faster.
>
> > Experiments are only conducted on small datasets. Can DensEMANN be applied on large datasets, for example ImageNet-1k?
>
> We are currently running experiments in this direction. See the global rebuttal.
>
>
> > Too many hyper-parameters are introduced in this method, which brings difficulties for applying this method on other tasks.
>
> We are currently testing DensEMANN on diverse tasks, ideally with few-to-no modifications to the hyperparameters. See the global rebuttal.
>
>
> > By automatically growing small DenseNet architectures, what insights can we obtain in architectural design?
>
> Two main insights:
> * Small, densely-connected architectures can reach high and even competitive performance on benchmark tasks. This further highlights how modern CNN models are often needlessly overparameterized.
> * There seems to be a limit to what one can do with DenseNet. The accuracy vs. size scatter plot in Figure 3 (at the end of Appendix A) suggests that there is a suboptimal Pareto front that is unique to small DenseNet topologies. This suboptimal front may be due to too many redundant elements in DenseNet architectures, as noted by previous works such as Log-DenseNet (Hu et al. 2017).
>
> The above insights open up the question of how far we can push the state of the art's collective Pareto front for accuracy vs. size. One way to do this is by making DensEMANN's search space broader and less restricted (see our answer to the question on GO methods)

---

> > ### Comment · Area_Chair_uuBo · 2023-08-20
> > **To Reviewer Jgdu: Please respond to the author rebuttal**
> >
> > Dear Reviewer Jgdu,
> >
> > The deadline for author discussion period is approaching soon. Please respond to the author's rebuttal and indicate whether your concerns have been addressed. Thank you!
> >
> > -AC

---

### Official Review · Reviewer_kBNK · 2023-07-05

**Soundness:** 3 good
**Presentation:** 4 excellent
**Contribution:** 1 poor
**Rating:** 3
**Confidence:** 3

**Summary:**

The authors study an algorithm for neural architecture search (NAS) called DensEMANN, which uses a progressive adaptation of a DenseNet architecture during training to find an efficient neural network for the target task.

**Strengths:**

The paper is very well written and the authors do a good job of explaining how the DensEMANN algorithm works.

**Weaknesses:**

I’m not sure that the paper currently has enough substance for publication. The authors’ spend the first 5 pages on introduction and description of the previously published DensEMANN algorithm. The changes made to this algorithm are described in section 3.2 in only 20 lines of text and appear to be primarily changes to the various hyperparameters of the existing algorithm.

The primary contribution of the paper is comparison of DensEMANN with other NAS methods on CIFAR-10 in section 4.3. The results are certainly interesting, but I think the authors should focus on quality per unit time rather than quality and parameter counts plotted in Figure 2. Plotting quality against the execution times in table 2 would make it much easier to compare DenseMANN to existing methods in efficiency, which is the primary property of interest, I think.

That being said, I’m not sure empirical comparison of an existing method with state-of-the-art methods is enough novelty to merit publication at NeurIPS. I’d encourage the authors to continue to develop their exploration. For example, clearly establishing a new state-of-the-art in efficiency for NAS. Or, taking the models from DensEMANN and studying their efficiency for deployment.

**Questions:**

I do not have specific questions aside from those listed above.

**Limitations:**

I did not identify potential negative societal impact of this work.

---

> ### Author Rebuttal · Authors · 2023-08-10
>
> > The primary contribution of the paper is comparison of DensEMANN with other NAS methods on CIFAR-10 in section 4.3. The results are certainly interesting, but I think the authors should focus on quality per unit time rather than quality and parameter counts plotted in Figure 2. Plotting quality against the execution times in table 2 would make it much easier to compare DenseMANN to existing methods in efficiency, which is the primary property of interest, I think.
>
> In the global rebuttal's enclosed figures PDF, we provide a scatter plot for error rate on CIFAR-10 vs. execution time in GPU days.
>
> > That being said, I’m not sure empirical comparison of an existing method with state-of-the-art methods is enough novelty to merit publication at NeurIPS. I’d encourage the authors to continue to develop their exploration. For example, clearly establishing a new state-of-the-art in efficiency for NAS. Or, taking the models from DensEMANN and studying their efficiency for deployment.
>
> We are currently exploring the application of DensEMANN to diverse datasets and application fields, and we aim to consider different kinds of NN architectures in future work. See the global rebuttal.

---

> > ### Comment · Reviewer_kBNK · 2023-08-17
> > **Reviewer response**
> >
> > Thank you to the authors for their response. I think that the additional directions for exploration you've listed are very interesting and would encourage the authors to pursue them. However, I don't think the additional results with CIFAR-100 are enough for me to raise my score.

---

> > > ### Author Response · Authors · 2023-08-20
> > >
> > > Dear Reviewer kBNK,
> > >
> > > Please excuse my late reply. I have just finished submitting my PhD thesis manuscript, which has kept me very busy during the last few days.
> > >
> > > Thank you very much for your reply. The fact that a NeurIPS reviewer says that our research route is "very interesting", although our results don't reach the level yet, is a strong motivation to keep up our future work and research.

---

### Official Review · Reviewer_aDQ3 · 2023-07-07

**Soundness:** 3 good
**Presentation:** 2 fair
**Contribution:** 2 fair
**Rating:** 5
**Confidence:** 3

**Summary:**

The paper presents  a new version of DensEMANN, an algorithm for generating small and competitive DenseNet architectures with optimal weight values. The authors aim to approach state-of-the-art performance for well-known benchmarks, or at least the state-of-the-art Pareto front between performance and model size. They achieve this by introducing a new version of the algorithm that uses a combination of layer pruning and weight optimization techniques. The authors evaluate DensEMANN on three popular image classification benchmarks (CIFAR-10, Fashion-MNIST, and SVHN) and show that it outperforms or matches the state-of-the-art methods in terms of accuracy and model size. The contributions of the paper are a new algorithm for generating small and competitive DenseNet architectures, a combination of layer pruning and weight optimization techniques, and state-of-the-art results on popular image classification benchmarks.

**Strengths:**

- The pruning and recovery stages which are the contributions of this paper are very well motivated and clearly explained. The way the pruning and recovery stages are designed is novel and using it in this context is rather unique.
- The authors have very aptly identified how DensEMANN fits in and very well introduced incremental approaches and NAS architectures.

**Weaknesses:**

- The difference in this paper with the original DenseEMANN is clearly communicated by the authors however all the the points mentioned except 2 (d):

> The pruning and recovery stages have been heavily modified to avoid long recovery
stages and their effects. We indeed observed that the kCS of settled filters is not
constant but actually decreases very slowly over time. If the recovery stage is too long,
this causes a very harsh pruning after which the accuracy cannot be recovered.

are just based on observation or not introduced in the paper or are very straightforward changes, I would suggest to consider only 2 (d) as a contribution of this paper.
-There are other aspects of this model which are very well framed and novel however it is important to note that these parts of the DensEMANN architecture arer not introduced in this paper but the original DensEMANN paper which reduces the novelty of this work by a huge margin.
- The paper does not provide a clear explanation of how the parameter limit of 500k was chosen for the experiments. Does going above these number of parameters make DensEMANN very computationally intensive or is unable to grow the network sensibly especially considering that 500k parameters in modern comparison are very few parameters especially for vision tasks?


**Questions:**

- This  did not affect my rating at all, but I do agree with this, I would hope that the authors could add a bit more in these statements to explain the significance of growing-based NAS algorithms:

> The most serious competitor to growing-based NAS algorithms are trainless or zero-cost algorithms
[30 , 31 , 2 ]. These evaluate candidate NN on basis of their performance with random weights. Such
methods can explore large search spaces in a matter of minutes or even seconds [31, 2]. However,
extra time is still needed for training the final candidate architecture in order to use it.

- The authors evaluate DensEMANN on three popular image classification benchmarks. Can they provide more insights into how the algorithm performs on other datasets (especially more complex larger datasets) or tasks, such as object detection or semantic segmentation (I think DensEMANN should very simply be able to approach this)? The datasets question is also a suggestion to include more experiments on that since the dataset complexity is a very important aspect in judging the efficacy of DensEMANN. However, as of the other tasks question, I was just wondering if the authors had already tried it out and faced any problems?
- A very common question from this paper is if the authors had tried replicating similar techniques for other larger similar architectures and for more modern architectures. Understanding if DensEMANN is able to generate more complex models is a very big question (and limitation)?

**Limitations:**

- With the current set of evaluation the authors do it is very hard to determine if DensEMANN like techniques can be applied for larger and more modern models

---

> ### Author Rebuttal · Authors · 2023-08-10
>
> > The paper does not provide a clear explanation of how the parameter limit of 500k was chosen for the experiments. Does going above these number of parameters make DensEMANN very computationally intensive or is unable to grow the network sensibly especially considering that 500k parameters in modern comparison are very few parameters especially for vision tasks?
>
> DensEMANN does not explicitly limit the number of parameters to 500k, or any other value for that matter. Rather, it implicitly limits the architecture's size through a core philosophy of "only growing what's strictly necessary" for a significant improvement in accuracy. This core philosophy is especially visible in the macro-algorithm's accuracy-based improvement criterion, which undoes the last layer's addition if it did not cause a significant change in the NN's accuracy. The micro-algorithm's pruning-recovery loops also contribute to limiting the size of each new layer to a bare minimum.
>
>
> > This did not affect my rating at all, but I do agree with this, I would hope that the authors could add a bit more in these statements to explain the significance of growing-based NAS algorithms
>
> Various authors in the past have characterized NAS as a bilevel problem, consisting of a parameter optimization problem nested into an architecture optimization problem (see e.g. "A Survey on Evolutionary Construction of Deep Neural Networks", Zhou et al. 2021).
>
> In growing and/or pruning algorithms, the aim is to parallelize these two levels as much as possible, optimizing the neural architecture and its weights at the same time. Meanwhile, in zero-cost approaches, the goal is to serialize the two levels, and to postpone the most computation-heavy of these two components (the training of candidate architectures) as much as possible. This said, as pointed out by White et al. (2022) in their ICLR blog post "A Deeper Look at Zero-Cost Proxies for Lightweight NAS", even if one uses zero-cost training proxies to optimize the NN's architecture, one still needs to optimize the architecture's weights (i.e. train the network) in order to achieve optimal performance on a target task.
>
> Furthermore, when used on their own for performance prediction, known zero-cost training proxies have got multiple disadvantages such as unreliable performance on different target tasks, and inherent biases towards certain topology patterns (again, see the blog post by White et al. 2022). Growing and pruning-based NAS algorithms can avoid these disadvantages in a simple and efficient way: by simultaneously training the network on the target dataset while suggesting and trying out changes to its neural architecture.
>
>
> > The authors evaluate DensEMANN on three popular image classification benchmarks. Can they provide more insights into how the algorithm performs on other datasets (especially more complex larger datasets) or tasks, such as object detection or semantic segmentation (I think DensEMANN should very simply be able to approach this)? The datasets question is also a suggestion to include more experiments on that since the dataset complexity is a very important aspect in judging the efficacy of DensEMANN. However, as of the other tasks question, I was just wondering if the authors had already tried it out and faced any problems?
>
> We are currently running experiments on larger and more complex datasets (in particular ImageNet1k), different kinds of tasks and application fields. See the global rebuttal.
>
>
> > A very common question from this paper is if the authors had tried replicating similar techniques for other larger similar architectures and for more modern architectures. Understanding if DensEMANN is able to generate more complex models is a very big question (and limitation)?
>
> This is one of our main planned research routes for future work (see the global rebuttal). We do believe that DensEMANN-based approaches are not limited to DenseNet, or even CNN. We are in particular interested in using a (Dens)EMANN-like method for growing RNN and Transformer networks.

---

> > ### Comment · Reviewer_aDQ3 · 2023-08-16
> > **Response to the authors rebuttal**
> >
> > > DensEMANN does not explicitly limit the number of parameters to 500k, or any other value for that matter. Rather, it implicitly limits the architecture's size through a core philosophy of "only growing what's strictly necessary" for a significant improvement in accuracy. This core philosophy is especially visible in the macro-algorithm's accuracy-based improvement criterion, which undoes the last layer's addition if it did not cause a significant change in the NN's accuracy. The micro-algorithm's pruning-recovery loops also contribute to limiting the size of each new layer to a bare minimum.
> >
> > I see, that makes sense, although I was wondering if 500k is the number of parameters after which DensEMANN is unable to generate useful parameters for the architectures and tasks you describe? For the lesser number of parameters, the results are impressive and I'm not undermining that but I was just wondering about this.
> >
> > > Various authors in the past have characterized NAS as a bilevel problem, consisting of a parameter optimization problem nested into an architecture optimization problem (see e.g. "A Survey on Evolutionary Construction of Deep Neural Networks", Zhou et al. 2021).
> >
> > Thanks, in my original review I was meaning if you could probably add this to the paper itself (a mere suggestion), which I think would be very helpful for readers to better understand the significance of the problem you pose.
> >
> > > We are currently running experiments on larger and more complex datasets (in particular ImageNet1k), different kinds of tasks and application fields. See the global rebuttal.
> >
> > Thanks for including and answering multiple of my questions in the global rebuttal especially about other architectures and datasets and tasks, as well as sharing a comparision. I do understand that running experiments on large datasets can be a bit difficult given the short rebuttal period.
> >
> > - For the CIFAR-100 experiments you do not limit the number of parameters and just let the network grow, right?
> > - As for the comparisons and experiments, it seems that after a certain number of parameters, DensEMANN is not able to generate parameters over some certain limit and thus just increasing execution time like for other architectures does not lead to considerable improvements. Mainly because the DenseNet trained for CIFAR-10 shown in the original paper has 25.6 M parameters, seeing DensEMANN generates very few parameters (<500k) after which it seems based on your rebuttal that it is unable to grow it sensibly?
> > - I was wondering if there was an explanation for why DensEMANN-generated models do not work really well for CIFAR-10? (I am not fixated on this paper beating SoTA but an explanation for these results would be very helpful)

---

> > > ### Author Response · Authors · 2023-08-20
> > >
> > > Dear reviewer aDQ3,
> > >
> > > Please excuse my late reply. I have just finished submitting my PhD thesis manuscript, which has kept me very busy during the last few days.
> > >
> > > > I see, that makes sense, although I was wondering if 500k is the number of parameters after which DensEMANN is unable to generate useful parameters for the architectures and tasks you describe? For the lesser number of parameters, the results are impressive and I'm not undermining that but I was just wondering about this.
> > >
> > > Yes indeed, in the sense that any newly grown architecture elements do not bring a significant improvement to the accuracy (as established by the macro-algorithm's improvement threshold).
> > >
> > > > Thanks, in my original review I was meaning if you could probably add this to the paper itself (a mere suggestion), which I think would be very helpful for readers to better understand the significance of the problem you pose.
> > >
> > > Thanks a lot for this suggestion! We will indeed add this to the paper.
> > >
> > > > Thanks for including and answering multiple of my questions in the global rebuttal especially about other architectures and datasets and tasks, as well as sharing a comparision. I do understand that running experiments on large datasets can be a bit difficult given the short rebuttal period.
> > > > * For the CIFAR-100 experiments you do not limit the number of parameters and just let the network grow, right?
> > >
> > > Indeed, as for all other datasets.
> > >
> > > > * As for the comparisons and experiments, it seems that after a certain number of parameters, DensEMANN is not able to generate parameters over some certain limit and thus just increasing execution time like for other architectures does not lead to considerable improvements. Mainly because the DenseNet trained for CIFAR-10 shown in the original paper has 25.6 M parameters, seeing DensEMANN generates very few parameters (<500k) after which it seems based on your rebuttal that it is unable to grow it sensibly?
> > >
> > > As explained above, whether an improvement is "significant" or not depends on the improvement threshold. By default, we set it to IT=0.01 (i.e. an improvement of 1 percentage point). In some settings, this may be seen as too strict of a limitation. If a lower IT value is set, then the algorithm will also accept more gradual improvements in the accuracy, and in that case perhaps the final DenseNet would reach various millions of parameters.
> > >
> > > This can be compared to the different DenseNet-BC architectures proposed in the original paper. As the number parameters was made to grow linearly at a constant rate (0.8M -> 15.3M -> 25.6M parameters), the error percentage decreased at an increasingly smaller rate (4.51% -> 3.62% -> 3.46%). Thus, adding the first 15M parameters causes an increase of almost 1 percentage point in accuracy, but adding 15M more parameters only causes an increase of 0.16 percentage points.
> > >
> > > > * I was wondering if there was an explanation for why DensEMANN-generated models do not work really well for CIFAR-10? (I am not fixated on this paper beating SoTA but an explanation for these results would be very helpful)
> > >
> > > Some hypotheses:
> > > * If the improvement threshold was set to a lower value, the accuracy may improve. This said, the parameter count would also increase...
> > > * In Annex A.3, we consider the possibility that DensEMANN's search space is too limited, and will always be bound to a suboptimal accuracy-vs-size Pareto front. Relaxing the search space's constraints (e.g. adding a broader choice of block-level configurations, some sparsity to the network's connections, or different kernel sizes) could enable the algorithm to reach more optimal results.
> > > * The training process could be fine-tuned. Taking the bilevel nature of NAS into account:
> > >   * Further regularization or data augmentation techniques could be used for training the learnable weights.
> > >   * Different growing-then-pruning operations (like our improvement and pruning stages) may be scheduled **solely to improve the learning process**. We are currently considering the possibility that, as in biological neural networks, **a quick and large growth** in neural capacity followed by **a harsh but judiciously chosen pruning** may result in improved learning capacity: the growing phase increases the network's pattern representation power, and the pruning phase keeps only the "essential" elements of the learnt patterns.

---

### Official Review · Reviewer_JTZn · 2023-07-08

**Soundness:** 2 fair
**Presentation:** 2 fair
**Contribution:** 2 fair
**Rating:** 4
**Confidence:** 2

**Summary:**

This paper proposes a new version of the existing DensEMANN, which grows small DenseNet architectures and trains them on target data. It claims that this version can quickly and efficiently search for small and competitive DenseNet architectures. The proposed approach has been evaluated on a number of benchmarks.

**Strengths:**

- The idea of automatically generate efficient architectures from a reference makes sense, and can be of great interests in many application scenarios.
- The proposed approach grows the architecture at both macro and micro levels, which seems to be a valid strategy.
- The proposed approach has been evaluated on various benchmarks, showing comparable performance with the state of the art.

**Weaknesses:**

- The delta compared to the original algorithm seems to be not very significant.
- The claim on being able to generate efficient densenet architectures is only supported by the number of parameters. For densenet like models the number of parameters might not be a good indicator for efficiency, due to many skip connections. Thus it seems not a very fair comparison.
- It is not clear how the proposed approach performs on larger datasets such as ImageNet.

**Questions:**

- How this approach performs on Imagenet?
- How the FLOPs/latency of the discovered models comparing with state of the art?

---

> ### Author Rebuttal · Authors · 2023-08-10
>
> > How this approach performs on Imagenet?
>
> We are currently testing DensEMANN on ImageNet1k. See the global rebuttal.
>
> > How the FLOPs/latency of the discovered models comparing with state of the art?
>
> In the below table we report the latency (in MFLOPs) of our discovered models for different datasets (CIFAR-10, Fashion-MNIST, SVHN, CIFAR-100) and settings of DensEMANN (with and without block replication, with and without CutOut regularization). The default configuration is "w/ repl. + cutout", i.e. with block replication and CutOut.
>
> | Dataset | DensEMANN setting | Latency (MFLOPs) |
> |:---:|:---:|:---:|
> | CIFAR-10 | w/o all | 56.57 ± 9.43 |
> |  | w/ cutout | 53.47 ± 5.67 |
> |  | w/ repl. | 74.81 ± 16.36 |
> |  | w/ repl. + cutout | 78.35 ± 22.85 |
> | Fashion-MNIST | w/o all | 8.39 ± 2.34 |
> |  | w/ cutout | 11.75 ± 7.15 |
> |  | w/ repl. | 11.79 ± 4.98 |
> |  | w/ repl. + cutout | 17.41 ± 8.94 |
> | SVHN | w/o all | 80.67 ± 10.24 |
> |  | w/ cutout | 73.91 ± 35.17 |
> |  | w/ repl. | 90.15 ± 43.86 |
> |  | w/ repl. + cutout | 139.75 ± 24.43 |
> | CIFAR-100 (IT=0.01) | w/ repl. + cutout | 105.05 ± 19.23 |
> | CIFAR-100 (IT=0.005) | w/ repl. + cutout | 156.31 ± 15.87 |
>
> N.B.: The full results for the CIFAR-100 architectures (including latency) are reported in the global rebuttal.
>
> In the below table we report the latency of the DenseNet-BC generated with the naive NAS baseline from Appendix A.2:
>
> | Dataset | Naive NAS setting | Latency (MFLOPs) |
> |:---:|:---:|:---:|
> | CIFAR-10 | N = 3, MPC = None | 92.54 ± 12.36 |
> |  | N = 3, MPC =   200k | 75.45 ± 0.00 |
> |  | N = 1, MPC =   None | 84.51 ± 19.08 |
> |  | N = 1, MPC =   60k | 61.01 ± 0.00 |
>
> For a comparison with a similar NN in the state of the art, according to the PyTorch docs, EfficientNet-B0 (Tan and Le 2020) has got a latency of 390 MFLOPS, and contains 5.29M trainable parameters. Its authors claim that it reaches 98.1% accuracy on CIFAR-10, but they give it a smaller size of 4M parameters.
>
> Also according to the PyTorch docs, MobileNet v2 (Sandler et al. 2019) has got a latency of 300 MFLOPs and contains 3.50M trainable parameters. MobileNet v3 (Howard et al. 2019) has got a latency of 60 MFLOPs, which is closer to our results, but with 2.54M parameters it is still much larger. As for ShuffleNet v2 0.5×, it has got a latency of 40 MFLOPs and 1.37M parameters. We believe that studying these architectures will be beneficial for reducing the latency of our auto-generated models.

---

> > ### Comment · Area_Chair_uuBo · 2023-08-20
> > **To Reviewer JTZn: Please respond to the author rebuttal**
> >
> > Dear ReviewerJTZn,
> >
> > The deadline for author discussion period is approaching soon. Please respond to the author's rebuttal and indicate whether your concerns have been addressed. Thank you!
> >
> > -AC

---

### Official Review · Reviewer_o4mm · 2023-07-25

**Soundness:** 3 good
**Presentation:** 3 good
**Contribution:** 1 poor
**Rating:** 4
**Confidence:** 2

**Summary:**

This paper proposed a new and improved algorithm to grow small DenseNet architecture from scratch while simultaneously training them from target data.


**Strengths:**

1. The paper is very clear and readable.
2. The evaluation is comprehensive and detailed, demonstrating the effectiveness.

**Weaknesses:**

1. The novelty is limited. The algorithm is backboned on a well-known algorithm, and the change to it is limited.
2. The scope of this algorithm is limited too.

**Questions:**

Please analyze what improvement can be brought by the differences in section 3.2.

**Limitations:**

Can this algorithm is adapted to other application fields?

---

> ### Author Rebuttal · Authors · 2023-08-10
>
> > Please analyze what improvement can be brought by the differences in section 3.2.
>
> 1. Changes to the macro-algorithm:
>     * (a) The macro-algorithm's last layer addition can always be removed because it is in fact always useless–at least from an accuracy improvement point of view. Indeed, the macro-algorithm only keeps adding layers if the latest layer addition has caused a significant change in the NN's accuracy. Otherwise, it stops. Keeping the last layer addition, which didn't change accuracy significantly, would go against DensEMANN's core philosophy of keeping those architecture elements that are necessary for a significant accuracy improvement (cfr. the abstract).
>     * (b) In the original DensEMANN paper (García-Díaz and Bersini 2021, Section IV.B), various observations were used for arguing against setting the improvement threshold (IT) to 0.005:
>         1. With IT = 0.005, the last few layer additions often only brought a very gradual increase in accuracy (or no increase at all), at the cost of more trainable parameters and a deeper architecture.
>         2. In these last few layers, the micro-algorithm pruned many of the generated filters, suggesting that the limited accuracy gains with these layers are only due to a few of their components.
>
>       For this reason, we decided to increase the default IT value to 0.01. Nevertheless, acknowledging that the observations in (García-Díaz and Bersini 2021) were limited (only 2 tests were performed for each dataset, and one of the tests crashed before completion), we tested an IT value of 0.005 again for the CIFAR-100 benchmark (see global rebuttal for results).
> 2. Changes to the micro-algorithm:
>     * (a) As explained in (García-Díaz and Bersini 2021, Section IV.A), when the "useful" and "useless" filter categories are made completely independent from the "settled" category, an undesirable phenomenon takes place: during the micro-algorithm's improvement stage, many new filters are detected as "useful" at the moment of their addition... but later become "useless" once they settle.
>
>       Consequently, due to the micro-algorithm's behaviour during the improvement stage (when a filter becomes useful a new filter is added), a great number of superfluous filters are added one-by-one very quickly. The result is an extremely overparametrized layer, which takes lots of time and computation resources to train. Afterwards, when these superfluous filters start settling and many of them turn out useless, they are pruned en-masse in the next pruning stage.
>
>       In conclusion, if non-settled filters can be counted as "useful", then multiple superfluous filter additions take place during the micro-algorithm's improvement stage, causing the NN's last layer–and DensEMANN's time and computation cost–to grow needlessly during this stage.
>
>     * (b) The smaller default value for the patience parameter (PP) was motivated by Fig. 5 of the original DensEMANN paper (García-Díaz and Bersini 2021, Section III.B). This figure shows the evolution of the kCS values for filters in minimal DenseNets, when these are trained on the CIFAR-10 dataset. It shows that, if a filter is created in mid-training, it takes around 40 training epochs (with a constant learning rate) for that filter's kCS to acquire a relatively stable value, i.e., to "settle" on a learnt operation. This change was also motivated by the quick rate at which other NAS algorithms similar to DensEMANN perform growing operations. For instance, NASH (Elsken et al. 2017) modifies the NN every 17 epochs.
>     * (c) In (García-Díaz and Bersini 2021), with PP=300, the main goal of the patience countdown was to impose a maximum period of time during which growing operations were allowed. In this paper, with a smaller PP value of 40, if growing operations are limited by the countdown then the algorithm may miss on important improvement opportunities. The goal of the countdown is thus different: it imposes a minimum training period before the pruning stage.
>     * (d) Observations in (García-Díaz and Bersini 2021, see Fig. 5) and our own preliminary tests showed that the kCS of settled filters does not actually remain constant, but in fact decreases slowly over time. After trying the original paper's DensEMANN configuration, we discovered that very long recovery stages often caused a very harsh pruning of the layer's filters, after which it was near-impossible for the NN to recover its prepruning accuracy. If the NN did manage to recover its accuracy, it was only after a very long recovery stage, which caused the vicious cycle to repeat. We thus had to implement a series of mechanisms to avoid long recovery stages and their negative effects.
> 3. Dense block replication: this feature was motivated by cell-based NAS approaches in other algorithms, such as NASNet (Zoph et al. 2018), ENAS (Pham et al. 2018) and DARTS (Liu et al. 2019). Our main aim was to verify if a significant increase in accuracy could be achieved by merely replicating the generated dense-block a user-defined number of times. The experiment was successful (there indeed was an increase in accuracy), and we plan to further study the implications of this in future work. In particular, due to the known limitations of purely cell-based approaches (see Fair DARTS, Chu et al. 2020) we are currently interested in a hierarchical NAS approach that searches for the right macro-structure along which to copy the dense blocks.
>
>
> > Can this algorithm is adapted to other application fields?
>
> We are currently working on this as our main priority. We are testing DensEMANN on multiple classification datasets from different application fields, and extending the algorithm to non-classification-based tasks (e.g. object detection and image segmentation). See the global rebuttal.

---

> > ### Comment · Area_Chair_uuBo · 2023-08-20
> > **To Reviewer o4mm: Please respond to the author rebuttal**
> >
> > Dear Reviewer o4mm,
> >
> > The deadline for author discussion period is approaching soon. Please respond to the author's rebuttal and indicate whether your concerns have been addressed. Thank you!
> >
> > -AC

---

> > ### Comment · Reviewer_o4mm · 2023-08-20
> >
> > Thank you for your reply! I think the paper writing can get improved. Maybe another storyline that emphasizes your idea is much better.

---

> > > ### Author Response · Authors · 2023-08-20
> > >
> > > Dear Reviewer o4mm,
> > >
> > > Tank you very much for your reply!
> > >
> > > I completely agree with it. This is a usual problem with my writing: I don't know how to "sell" my ideas well.
> > >
> > > Other people have already pointed it out to me, and I will make my best to improve on my storytelling in the future.

---

### Author Rebuttal · Authors · 2023-08-09

### Concerning larger datasets (in particular ImageNet1k):
* We have run extra tests on CIFAR-100.

  The dataset split was identical to that of CIFAR-10: a training set of 45,000 random "training" images, a validation set of 5,000 random "training" images not already in the training set, a test set consisting of all 10,000 "test" images in the original dataset.

  The preprocessing was identical to that of CIFAR-10: random crop + horizontal flip, normalization (with the same mean and SD values as CIFAR-10), cutout regularization.

  DensEMANN's configuration is the same as for all other tests in the main paper, although we also tried to set the improvement threshold to IT = 0.005 to see if it affected performance significantly.

  The time measurements for IT = 0.01 were all taken *sequentially* (i.e. one test at a time) on an MSi GT76 laptop. For IT = 0.005, they were performed *in parallel* in our internal cluster (up to three tests at the same time), and so we consider them to be less reliable. (See the paper for the full specs.)

  The results (mean and standard deviation over 5 runs) are shown in the tables below:

| Dataset | GPU execution time (hours) | GPU inference time (seconds) | Num. layers per block | Trainable parameters (k) | Latency (M FLOPs) |
|:---:|:---:|:---:|:---:|:---:|:---:|
| CIFAR-100 (IT=0.01) | 15.41 ± 2.34 | 3.52 ± 0.32 | 7.6 ± 1.7 | 269.72 ± 48.42 | 105.05 ± 19.23 |
| CIFAR-100 (IT=0.005) | 22.10 ± 1.94 | 5.99 ± 0.70 | 11.0 ± 1.2 | 402.68 ± 41.89 | 156.31 ± 15.87 |

| Dataset | Validation set acc. (%) | Validation set loss | Test set acc. (%) | Test set loss |
|:---:|:---:|:---:|:---:|:---:|
| CIFAR-100 (IT=0.01) | 68.09 ± 0.33 | 1.16 ± 0.02 | 72.43 ± 0.60 | 1.05 ± 0.05 |
| CIFAR-100 (IT=0.005) | 68.54 ± 0.78 | 1.12 ± 0.02 | 73.50 ± 0.61 | 0.98 ± 0.01 |

* We are currently running tests on ImageNet1k. These will take some time: each epoch lasts around 8-12 minutes, so a full run of DensEMANN will take at least half a month. We can run various tests in parallel though, but our computing resources are limited to what is described in the paper (MSi GT76, MSi GT75, internal cluster).

### Concerning other problems and application fields:
* We are planning to try DensEMANN on the ESC-50 audio dataset. The data will be turned into spectrograms, so CNN like DenseNet can still be applied to infer the labels.
* Similarly, we may use DensEMANN on other data that can be represented in a spectrogram-like manner, such as the electromyography wave signals in NinaPro DB5.
* We are also planning to try DensEMANN on object detection or image segmentation problems. We may still use DenseNet for this, but since the output channels and accuracy measures for these problems are different to those for classification this will take some more time to code. We may also develop a new version of the algorithm that grows architectures specifically designed for object detection (see below).

### Concerning other NN architectures:
* This is one of our main planned research routes for future work.
* We wish to try (Dens)EMANN-like approaches for the following kinds of NN:
    * U-Nets (for image segmentation).
    * YOLO v4-v6 like architectures (for object detection).
    * Generative adversarial networks (GANs).
    * Recurrent neural networks.
    * Transformer networks.
* One or various of the above research routes will be followed in a future paper.

### Comparison with the state of the art:
At the request of one of the reviewers, in the enclosed PDF we provide a new scatter plot that compares DensEMANN to the NAS algorithms in Table 7, this time in terms of error rate on CIFAR-10 vs. algorithm execution time in GPU days.

---

### Comment · Area_Chair_uuBo · 2023-08-18
**To Reviewers: Please respond to the author rebuttals.**

Dear reviewers,

Thank you for serving as a reviewer for NeurIPS!

We are towards the end of the discussion stage with authors, but some of you haven't posted your response to the author rebuttals yet.
As the scores for this paper are diverse, please check the author rebuttals, reply to them and update your score (when necessary) ASAP. Thanks!

-AC

---

### Decision · Program_Chairs · 2023-09-21

**Decision:**

Reject

**Comment:**

This paper introduces an improved version of DensEMANN, specifically tailored for the generation of compact DenseNet architectures. The method effectively combines pruning and weight optimization techniques to yield competitive results on benchmark datasets. However, reviewers have expressed reservations regarding the novelty and depth of this work. The extent of improvement over the original algorithm appears to be relatively modest. Additionally, reviewers have emphasized the need for a more explicit articulation of the paper's motivation, a thorough comparison with the original method, and an assessment of its scalability to larger datasets. In light of these concerns, the meta-reviewer deems the paper unsuitable for publication.